# The Effect of Material Thickness, Load Density, External Airflow, and Relative Humidity on the Drying Efficiency and Quality of EHD-Dried Apples

**DOI:** 10.3390/foods11182765

**Published:** 2022-09-08

**Authors:** Anjaly Paul, Alex Martynenko

**Affiliations:** Department of Engineering, Faculty of Agriculture, Dalhousie University, Truro, NS B2N 5E3, Canada

**Keywords:** electrohydrodynamic drying, apples, quality, multifactorial analysis, hot air drying

## Abstract

Electrohydrodynamic drying is a novel non-thermal technique for dehydrating heat-sensitive foods. However, its industrial applications are limited due to the underexplored effects of material properties and environmental conditions on product quality. For this purpose, a multifactorial experiment was designed to study the effects of material thickness, load density, external airflow, and humidity on the EHD drying efficiency and quality of apple slices. The experiments show that the intensity of EHD drying increased with a decrease in humidity, slice thickness, and load density. The effective diffusivity of apple slices with EHD drying was about 5.17·10^−12^ m^2^/s, slightly increasing with external airflow. The specific energy consumption of EHD drying was 10–12 times lower than hot air drying. The time of EHD drying at 20 °C was equivalent to hot air drying at 40 °C, but the impact of EHD drying on the product quality was significantly lower. EHD drying better preserved the color and phenolic content in dried apple slices, with less cellular damage. Hence, EHD drying can be employed in industry as a sustainable alternative to hot air drying.

## 1. Introduction

Electrohydrodynamic (EHD) drying is a new and promising non-thermal technique for the dehydration of heat-sensitive foods. It utilizes high voltage corona discharge between two electrodes (pin/wire to plate/mesh), resulting in air ionization and ionic wind [1,2]. The advantages of this drying technique, such as energy efficiency, superior product quality, and low capital/operational cost, have been proven in lab research [3] and on the industrial scale [4]. However, industrial applications of this technology are limited due to the underexplored effects of material properties and environmental conditions on the efficiency of EHD drying.

Most studies on EHD drying of foods are focused on the effects of electrical parameters and electrode configuration on drying efficiency and energy consumption [2,5]. Less attention has been paid to the effects of material properties and environmental conditions, such as air temperature, velocity, and relative humidity. The impact of material thickness on EHD drying was evaluated for potato slabs [5], fish [6], and beef jerky slices [7]. It was found that the drying rate decreased with an increase in thickness in all cases. In addition, EHD drying is more suitable for a thin layer of materials because of the convective moisture transfer mechanism. The effect of the load density of apple slices on the drying kinetics and specific energy consumption was explored by Onwude et al. [8]. They reported that the decrease in load density from 70% to 3% reduced the drying time but increased specific energy consumption. The effect of external airflow on EHD drying was more complex. The drying rate increased with air velocity to some critical values and then decreased for some non-food materials [9,10]. The critical air velocity depended on the electric field intensity [10]. In another study, Dinani et al. [11] observed the complex relationship between EHD voltage and external air velocity for mushrooms. The relative humidity (RH) is another crucial factor affecting EHD drying kinetics, but its effect is underexplored. Bai et al. [6] reported that low RH is beneficial for EHD drying of fish. Similar observations were made by Martynenko et al. [12] for EHD drying of mushroom slices. In a literature review by Zhang et al. [13] on the effect of RH on hot air drying of fruits and vegetables, it was reported that high RH could facilitate heat transfer and increase the drying rate in the constant rate period of drying. Whereas, low RH enhances moisture diffusion in the falling drying rate period. However, these conclusions, relevant to hot air drying, have never been tested for EHD drying.

Product quality is crucial for the industrial adaptation of the EHD drying technique. The quality of EHD-dried products was mostly compared with natural or hot air drying [3]. It was widely reported that EHD-dried products retained the natural quality better than hot air-dried products. EHD drying preserved pigments such as chlorophyll in spinach [14] and carotenoids in carrots [15] with the final color almost similar to the fresh fruits and vegetables [16,17,18,19]. The rehydration ratio was higher in EHD-dried potatoes, wolfberry, and radish [20,21,22] with a superior texture [23]. Furthermore, EHD drying of various fruits and vegetables such as spinach, quince, mushrooms, and carrots reported smaller shrinkage [11,17,22,24] with fewer microstructural changes in fruits such as wolfberry and apricots [25,26]. The bioactive and nutritional components such as vitamins [4,14], flavonoids [19], and polysaccharides [21] were also better preserved in EHD drying of fruits and vegetables. In most cases, the quality changes were correlated with EHD operating conditions such as voltage [27,28,29], external airflow [11,28,30], or the effect of pre-treatments [25,31]. Ni et al. [21] found that the cellular disintegration of Chinese wolfberry was proportional to the applied voltage and ionic wind intensity. Unfortunately, research on the effects of material properties and environmental factors on product quality is very scarce. A recent study reported a considerable effect of thickness on the shrinkage, color, and browning of EHD-dried apple slices [32]. Martynenko et al. [30] reported that external airflow below 1.0 m/s reduced the enzymatic browning in EHD drying of apple slices. However, external airflow above 1.0 m/s provoked significant visible browning [11,31].

The literature review shows that material properties and environmental conditions are among the factors affecting EHD drying kinetics. However, the influence of these factors on quality attributes, such as shrinkage, color, rehydration ratio, disintegration index, and total polyphenolic content, has not yet been investigated. A focused study on the combined effects of material thickness, load density, airflow, and humidity on the efficiency and quality of EHD drying is required to upscale this technology to industrial application. This challenge was addressed by a multifactorial experimental design, followed by statistical analysis.

## 2. Materials and Methods

### 2.1. Experimental Apparatus

A lab-scale EHD dryer was modified for the study [32]. The EHD system was equipped with the pins-to-mesh electrode system, as displayed in Figure 1. The discharge electrode was made of 72 stainless steel (SS) pins, arranged in a rectangular grid with 2 cm spacing on a fiberglass breadboard 240 × 170 × 2 mm (Vector Electronics Inc., Canada). All pins were electrically connected by stainless steel wire, which was connected to a high-voltage DC power supply (Universal Voltronics, Atlanta, IL, USA). The discharge electrode (230 × 160 mm) was insulated from the grounded frame by plastic rods screwed to the stationary plexiglass frame on the top of the holder.

The collecting electrode with the same dimensions (230 × 160 mm) was made of stainless-steel woven mesh (Dorstener Wire Tech, Spring, TX, USA) with a 0.406 mm thick wire and a 0.864 mm opening. The mesh was framed into a plastic enclosure, protecting it from mechanical deformation. An additional 0.66 mm thick plastic mesh (Aboat, China) was placed on the collecting electrode preventing the sticking of apple slices to the electrode. This assembly was supported by plastic rods with 4 cm height, connected to a rectangular plexiglass support to minimize the effect of the electric field on a sensitive digital balance. A height adjusting jack was used to precisely set the electrode gap between the tip of the discharge electrode and the surface of the dried material. The experiments with EHD drying were performed at 20 kV voltage and a 4 cm electrode gap, providing an electric field strength of 5 kV/cm. The EHD drying setup is portrayed in Figure 2.

The experimental setup consisted of electrodes, a digital scale, a computer, an air blower, and a high-voltage positive (+) DC source. The weight changes of the apple slices during drying were periodically (10 s) recorded on the computer using Adam DU software version 1.11.56 (Adam Equipment, Kingston, England). The external airflow in the drying chamber was controlled using a 1.5 kW blower (model 3HMJ5, Dayton Electric Co., Lake Forest, IL, USA) attached to a flexible aluminum duct. The rectangular outlet of the 260 × 100 mm duct was fixed 9 cm away from the electrodes. An airflow straightener, inserted in the duct outlet, prevented turbulence and uneven airflow distribution above the surface of dried material. The relative humidity (RH) in the drying room was measured using a wet and dry bulb hygrometer (model B6030, Baker Instruments, NY, USA). The temperature in the drying room was about 20 ± 2 °C.

### 2.2. Drying Experiment

Fresh apples (McIntosh) for the experiments were collected from a local farm and kept in refrigerated storage at 4 ± 1 °C until use. Apples were cut into square slices without peel and core as mentioned in Paul et al. [32]. The slices of varying thickness (1, 2, and 3 mm) were spread uniformly at 50% and 100% load densities on the mesh. At 100% load density, 56 apple slices were arranged in 6 × 9 rows without any gap between the slices, and for 50% load density, 27 slices were arranged in a chess pattern with a 1-inch gap, 4 and 5 slices in every alternate row. The experiments were conducted either without external airflow or at controlled air velocities of 0.5 and 1 m/s. The air velocity was confirmed with a hot wire anemometer (model 405i, Testo instruments, Canada). The apple slices were dried to the equilibrium moisture content. To evaluate the effectiveness of EHD drying, the quality of EHD-dried apple slices were compared with apple slices (2 mm) dried in a hot air dryer (Tray dryer, UOP 8A, Ringwood, England) at 40 °C and 0.15 m/s. It was not possible to control relative humidity in this experimental setup. The range of the variables used for the multifactorial experiment is shown in Table 1. A general factorial design with three factors (3 × 3 × 2) was created in Minitab 19 (Minitab LLC, Pennsylvania, USA), and the experiments were performed in triplicates.

Each set of experiments was carried out in random order and the results were statistically analyzed. The experiments with outliers were rejected and repeated.

### 2.3. Drying Characteristics

#### 2.3.1. Drying Kinetics

The moisture content (*M*) (g H_2_O /g dry weight) and moisture ratio (*MR*) of apple slices were determined from the weight changes:(1)$$M=\frac{W_{i}-W_{d}}{W_{d}}$$
(2)$$MR=\frac{M_{t}-M_{e}}{M_{o}-M_{e}}$$
where *W_i_* (g) and *W_d_* (g) denote the initial and dry weight of the apple slices. The dry weight was determined using the standard oven drying method at 105 °C for 24 h [33]. In Equation (2), *M_o_* refers to the initial moisture content, which is 8.7 ± 0.2 g/g. *M_t_* refers to the moisture content at the time *t,* and *M_e_* refers to the equilibrium moisture content. The final moisture content of completely dried apple slices at a particular relative humidity was used as the equilibrium moisture content and it was verified using the moisture sorption isotherm of apples at 20 °C [34]. Further, the drying rate (*DR*) of apple slices was determined by the weight change over time (Equation (3)).
(3)$$DR=\frac{W_{t}-W_{t+\Delta t}}{dt}$$
where *W_t_* is the weight of the sample at time *t*, and *dt* (s) is the time increment.

To normalize the impact of the area on the drying rate, we used the drying flux (*ṁ*), calculated from the given equation:(4)$$ṁ=\frac{1}{A}\times DR\;$$
where *ṁ* (g/m^2^s) is the drying flux, *A* (m^2^) is the area of evaporation, and *DR* is the average drying rate obtained from the slope of moisture ratio versus time in the constant rate period of drying. The drying flux was calculated for the initial 30 min in a constant drying rate period.

#### 2.3.2. Drying Rate Constant

The drying rate constant, *k* (h^−1^), was determined from the moisture ratio curves in the falling drying rate period (0.5 < *MR* < 0.1) using the exponential model for thin-layer drying:(5)$$MR=e^{-kt}$$
where *MR* is the moisture ratio and *t* (h) is the time.

#### 2.3.3. Effective Moisture Diffusivity

The effective moisture diffusivity (*D_eff_*) was evaluated from Fick’s second law in the falling rate period. The final thickness, *L* (m), was used for the calculations, as most of the shrinkage happens in the initial period of drying. The following equation was used for calculating *D_eff_*_,_ considering the apple slice as an infinite slab.
(6)$$MR=\frac{8}{\pi^{2}}\exp(\frac{-\pi^{2}D_{eff}t}{4L^{2}})$$
(7)$$D_{eff}=\frac{-4L^{2}}{\pi^{2}t}\ln\;(\frac{\pi^{2}}{8}MR)$$

#### 2.3.4. Specific Energy Consumption (SEC)

The *SEC* (kJ/kg) of EHD drying was calculated as:(8)$$SEC=\frac{V\times I}{DR}$$
where each term refers to the applied voltage *V* (kV), electric current *I* (mA), and drying rate *DR* (kg/s).

### 2.4. Quality Attributes

#### 2.4.1. Shrinkage

The initial (fresh) and final (dry) area and thickness of apple slices were determined through image analysis using the procedure explained by Paul et al. [32]. The area and volumetric shrinkage were calculated as:(9)$$A_{shrinkage}=\left(1-\frac{A_{f}}{A_{i}}\right)\times 100$$
(10)$$V_{shrinkage}=\left(1-\frac{V_{f}}{V_{i}}\right)\times 100$$
where *A_i_* and *A_f_* are the initial and final area, and *V_i_* and *V_f_* are the initial and final volume of apple slices.

#### 2.4.2. Color

Similarly, the color changes were also determined using image analysis [32]. The color change, ΔE, for each set was determined as:(11)$$\Delta E\;\sqrt{{\left(L^{*}-L_{o}^{*}\right)}^{2\;}+{\left(a^{*}-a_{o}^{*}\right)}^{2}+{\left(b^{*}-b_{o}^{*}\right)}^{2}}$$
where *L_o_**, *a_o_**, and *b_o_** are the initial color values of the apple slices and *L**, *a**, and *b** are the final color values. Similarly, the color change was measured for the hot air-dried slices.

#### 2.4.3. Rehydration Ratio

To assess the rehydration capacity of the EHD and hot air-dried apple slices, 2 g of the dried slices were immersed in 30 mL ultrafiltered water at 25 °C for 1 h. The water was drained, and the excess water was removed by lightly pressing the slices with absorbent paper. The rehydrated slices were weighed and the rehydration capacity was determined using the given equation:(12)$$RR=\frac{Wt-Wd}{Wd}\times 100$$
where *RR* (*%*) is rehydration ability; *W_t_* is the weight of rehydrated slices, g; and *W_d_* is the weight of the dried slice, g.

#### 2.4.4. Disintegration Index

The electrical conductivity disintegration index, *Z_p_*, was used for the estimation and comparison of damage on apple slices after EHD or hot air drying. The initial electrical conductivity, *σ_i_* (S/m), of fresh intact apple slices was measured using a benchtop conductivity meter (Model 860031, SPER Scientific, Scottsdale, AZ, USA). The slices were added to 30 mL distilled water and mechanically shaken in a rotary shaker for 30 min. Similarly, the electrical conductivity *σ* was measured after drying. Then, the conductivity of completely damaged slices, *σ_d_*, was determined after the drying of slices in a hot air oven at 105 °C for 24 h (dry matter), and the disintegration index can be calculated using the equation:(13)$$Z_{p}=\frac{\sigma -\sigma_{i}}{\sigma_{d}-\sigma_{i}}\;$$
where *Z_p_* is ‘0’ for intact apple slices and ‘1’ for completely damaged slices [21].

#### 2.4.5. Total Phenolic Content (TPC)

The TPC of EHD and hot air-dried apple slices was determined using the Folin–Ciocalteu method with slight modifications to the procedure described in Singleton et al. [35]. The dried apple slices were ground to powder, and 0.5 ± 0.01 g of powder was weighed and mixed in 10 mL ultrafiltered water. The polyphenols were extracted in a water bath at 100 °C for 30 min. The mixture was cooled to room temperature and centrifuged at 4500 rpm for 15 min. The supernatant (200 µL) was carefully pipetted to 1 mL of 1:10 diluted Folin’s reagent. After about 4 min, 800 µL of 7% sodium carbonate solution (Na_2_CO_3_, 75 g/L) was added to the solution and incubated in the dark at room temperature for 2 h. Gallic acid at concentrations of 0–500 mg/L was used to plot the standard calibration curve. The absorbance against the blank was determined at 765 nm using a UV visible spectrophotometer (model Genesys 150, Thermo Fisher Scientific, Waltham, MA, USA). The TPC of the dried slices was expressed in mg of gallic acid equivalents (GAE) per gram of dry weight [36].

### 2.5. Statistical Analysis

The multifactorial experiment was designed with four factors using a full factorial design in Minitab 19 software (Minitab LLC, Pennsylvania, USA). The analysis of variance (ANOVA) using a general linear model was performed to study the effect of thickness, load density, external airflow, and RH on the drying characteristics and quality. The assumptions of normality and constant variance were verified for each response variable by examining residuals. Tukey’s post hoc test was used for the multiple mean comparisons at a 0.05 significance level. Since relative humidity could not be controlled during the experiment, it was used as a covariate in the multifactorial analysis.

## 3. Results and Discussion

### 3.1. Drying Characteristics

A multifactorial experiment was conducted to evaluate the effect of slice thickness, slice load density, external airflow, and relative humidity (RH) on the drying kinetics of apple slices. The experiments were performed in random order with three replicates over 5 months (November 2021–March 2022). The initial moisture content of apple slices was 8.7 ± 0.2 g/g and it was dried to the equilibrium moisture content of 0.38 ± 0.02 g/g. The example of the drying kinetics of apple slices of different thicknesses is shown in Figure 3.

It follows that within the first 30 min of drying (1 < *MR* < 0.8), there were linear drying kinetics, which indicated a constant drying rate period. This behavior was observed in both EHD and hot air drying. Hence, the drying rate was determined as the slope of weight changes (g) versus time (s). To exclude the effect of load density (evaporation area), the drying rate (g/s) was divided by the total area (m^2^) to obtain the drying flux.

The effects of the above-mentioned factors on the drying flux, drying rate constant, effective diffusivity, and specific energy consumption of EHD drying in various combinations have been quantified and compared with hot air drying, in the following Section 3.1.1, Section 3.1.2, Section 3.1.3 and Section 3.1.4. The data are presented as combined effects of three factors, namely thickness, load density, and airflow velocity with the corresponding coding (Table 1).

#### 3.1.1. Drying Flux

In food drying, material thickness is a critical geometrical characteristic, determining the mode of mass transfer. It was found that EHD is more effective for thin-layer drying due to the convective mechanism of mass transfer [2]. Previous studies reported that the EHD drying rate was significantly reduced with the increased thickness of potato slabs [5] or beef jerky slices [7]. Our experiments (Figure 4) showed that the drying flux is independent of thickness. This could be explained by the convective action of EHD, which affects only surface water, depending on the surface area, but not thickness. On the other hand, the effect of the material load was significant. The drying flux was considerably higher at 50% load density, implying higher moisture transfer rates (Figure 4). This could be attributed to the better convection regime, where moisture escapes through the gaps between slices. This assumption is supported by the additional positive effect of forced airflow at 50% load, which is almost negligible at 100% load.

The general effect of airflow on the drying flux was more complex, depending on load density. This complexity is related to the possible interaction between external airflow and EHD flow [10]. In our experiments, the most consistent positive effect of external airflow was observed at a low air velocity of 0.5 m/s (Figure 4), which agrees with earlier observations [9]. A suppression effect of airflow at high velocities on the ionic wind was reported elsewhere. The possible explanation was that the external airflow suppresses the positive effect of EHD-induced airflow by blowing ionic wind away from the material surface. At low electric field strength, the ionic wind becomes weak, and even small external airflow could interrupt the ionic wind. The drying flux of hot air-dried apple slices was similar to EHD drying.

The relative humidity (RH) in the multifactorial experiments was variable in the range of 50–65% and therefore considered an uncontrollable factor. The single effect of RH on the drying flux was evaluated in a separate set of experiments (2 mm—100% load—0 m/s). The air temperature in the relative humidity trials was about 22 ± 4 °C.

Figure 5 shows the effect of RH on the intensity of EHD drying. In the range from 40 to 50% RH, the drying flux was constant, so this range does not affect the electro-convective mass transfer. As the RH increases, the drying flux of apple slices decreases. This could be explained by the decrease in the water vapor pressure gradient, which is the major driving force of drying [37]. Humid air loses the ability to absorb more water. Additionally, the formation of ionic hydrates at a higher RH could be an additional factor, decreasing the mobility of charged particles, which, in turn, reduces the drying efficiency [12,38,39]. It could be observed that relative humidity above 65% is considerably limiting for EHD drying.

#### 3.1.2. Drying Rate Constant

The drying rate constant *k* was determined as the slope of moisture ratio vs the time graph using Newton’s exponential model (R^2^ > 0.99) in the falling drying rate period (0.5 < *MR* < 0.1). Higher *k* values indicate a faster drying process. The drying rate constants from the multifactorial experiment are shown in Figure 6.

In our experiments, the *k* value decreased significantly with the increasing thickness, which could be explained by the drying theory. This was in agreement with the previous observations on EHD drying of fruits [32]. The increase in air velocities also resulted in a higher drying rate constant (Figure 6). Similar effects of external airflow on increasing drying rate constant were observed by Martynenko et al. [30] in EHD drying of apple slices. However, the slice load density did not affect the drying rate constant. A possible explanation is that at *MR* < 0.5 the shrinkage of slices created a gap between them for the moisture to escape. The highest drying rate constant in EHD drying was for 1 mm slices at 50% load density and 1 m/s airflow. The drying rate constant for hot air drying was notably higher, indicating the influence of temperature on diffusive moisture transfer in the falling drying rate period [40].

#### 3.1.3. Effective Diffusivity

The effective diffusivity (*D_eff_*) was calculated from Equation (7), taking into account the shrinkage of the sample. Various studies have investigated the influence of EHD drying on effective diffusivity. For instance, *D_eff_* increased with electric field strength for banana slices [41] and applied voltage for mushroom slices [42]. The study by Dinani et al. [42] revealed that the *D_eff_* in EHD-assisted hot air drying at 60 °C was 2.3 times higher than hot air drying alone. Ni et al. [25] reported that the chemical and ultrasound pre-treatments of goji berries before EHD drying significantly improved effective diffusivity. Figure 7 portrays the effective diffusivity of EHD and hot air-dried apple slices.

The effective moisture diffusivity increased considerably with an increase in air velocity (Figure 7). This concurs with the observations on the EHD drying of white champignons with forced air convection [12]. The effect of temperature was also significant. The effective diffusivity of EHD drying at 20 °C was (0.517 ± 0.11) × 10^−11^ m^2^/s, which was significantly lower than that of hot air drying at 40 °C (1.216 ± 0.14) × 10^−11^ m^2^/s. This difference indicates the major effect of temperature on moisture diffusion [40].

#### 3.1.4. Specific Energy Consumption

EHD drying is highly acclaimed for its low energy consumption [43]. The ratio of energy used for moisture evaporation to the energy supplied to the dryer is defined as the efficiency of a dryer. However, it may vary depending on the total energy consumption, which includes the energy consumed by the power supply and auxiliary equipment. The specific energy consumption (kJ/kg), determined from the voltage and current used by the EHD dryer, is a more accurate parameter in quantifying the energy required to evaporate a unit mass of water [12,44]. This removes the energy losses caused by individual parts in the dryer assembly.

In this study, the specific energy consumption (SEC) of EHD drying was in the range of 390.85 kJ/kg to 673.68 kJ/kg, which was 12–20 times lower compared to hot air drying (Figure 8). Huge differences in the energy consumption of EHD and hot air drying were reported by other researchers in the drying of carrots [17], mushrooms [12], and kiwi slices [16]. It was no difference, however, among EHD treatments. This again confirms the potential of EHD drying as a sustainable alternative to conventional hot air drying techniques.

#### 3.1.5. Statistical Analysis

Table 2 summarizes the statistical ANOVA *p*-values from the multifactorial experiment, showing the significance (*p* < 0.05) of the main and interaction effects of factors (external airflow, slice thickness, and slice load density) on the EHD drying of apple slices.

The ANOVA results from Table 2 reveal significant the main and interaction effects of the considered factors on the drying characteristics. There were significant interaction effects of thickness–load and load–airflow on the drying flux. It should be noted the highly significant effect of slice load density alone. This could be explained by the positive effect of load distribution on lateral diffusion, as well as convection mass transfer. The reason for the significant interaction effect (thickness*load) on the drying flux was not completely clear. This interaction is possibly related to the positive effect of spacing between slices on the convective mass transfer. The drying rate constant *k* was substantially affected by both thickness and airflow. The main effects of thickness and airflow reflect improved diffusive and convective mass transfer in the falling rate period of drying. Likewise, a considerable effect of external airflow on effective diffusivity was observed.

The significant effect of load density on SEC could be explained by the fact that drying of 100% or 50% requires the same amount of energy in kJ, but the amount of evaporated water (kg) is twice less. So, the efficiency of EHD drying with 50% load density is twice less than 100% load density and further decreases proportionally to the decrease in the load. This was similar to the results of Onwude et al. [8], where an increase in load density decreased the EHD energy efficiency. Hence, the optimization of material load density is important for the commercialization of EHD drying because even though decreased load density provides faster drying, it might not be economical.

### 3.2. Quality

The effects of EHD drying on quality parameters, such as shrinkage, color, rehydration ratio, disintegration index, and total phenolic content, are summarized in the following Section 3.2.1, Section 3.2.2, Section 3.2.3, Section 3.2.4 and Section 3.2.5 The data are presented as combined effects of three factors, namely thickness, load density, and airflow velocity with the corresponding coding (Table 1). The main and interaction effects of input factors on each quality attribute in comparison with hot air drying are discussed in detail.

#### 3.2.1. Shrinkage

Shrinkage occurs due to the microstructural stress and collapse of the cellular structure during drying [45]. Moisture removal results in the loss of turgor pressure and subsequent shrinkage [46]. In this study, the shrinkage was evaluated through sample imaging using a computer vision setup [47]. The area and volumetric shrinkage of EHD and hot air-dried apple slices are shown in Figure 9.

The area and volumetric shrinkage depended on the thickness of apple slices. From Figure 9A, it follows that 1 mm slices experienced slightly smaller shrinkage than 2 and 3 mm slices. This could be due to the shorter drying time, considering that shrinkage increases with drying time [48]. There was no significant effect of external air velocity and load density on the shrinkage of EHD-dried apple slices. The volumetric shrinkage of 82–92%, obtained in our study, was similar to the shrinkage values reported for EHD-dried mushrooms [11]. In their study, volumetric shrinkage increased with an increase in air velocity from 0.4 to 2.2 m/s. In contrast, in our study, the effect of airflow was not observed. This could be explained by the smaller air velocity, which did not exceed 1.0 m/s. The area shrinkage of apple slices was 36.21 ± 6.29%, lower than in hot air drying (46.78 ± 3.53%), indicating less microstructural stresses in EHD. This agrees with the results obtained in previous studies [17,24]. Surprisingly, the volumetric shrinkage did not show any significant difference between EHD and hot air drying (Figure 9B).

#### 3.2.2. Color

The color of foods affects the consumer’s perception of quality. Hot air drying usually affects color due to pigment degradation, browning, or oxidative reactions [49]. The enzymatic browning due to polyphenol oxidase enzymes is mainly responsible for the browning in apple slices in the presence of oxygen, while non-enzymatic browning occurs due to Maillard reactions at high temperatures [50,51]. Hence, non-thermal drying techniques focus on preserving the color and appearance of dried products. It is known that corona discharge produces a variety of ions, ozone, and other highly reactive species, which could inhibit enzymes and halt the browning reactions [52]. The color changes in the multifactorial EHD drying experiment were compared with hot air drying in Figure 10.

In our experiments, there was no significant difference in color among EHD treatments. The average color change (ΔE) was 1.53 ± 0.4 for EHD versus 6.78 ± 1.05 for hot air-dried slices (Figure 10). The smaller color changes in EHD-dried slices compared to hot air-dried slices could be attributed to enzyme inactivation, as well as the lower temperature of drying. Similar results were obtained in EHD drying of carrot slices [17], kiwi slices [16], and tomato slices [48]. Based on the results of our study, we can conclude that EHD drying is effective in reducing the browning reactions and maintaining the physical appearance of dried apple slices.

#### 3.2.3. Rehydration Ratio

The rehydration ratio is a crucial parameter used to determine the dried product quality. A higher rehydration ratio indicates lesser damage to the cellular structure and water-holding components such as protein and starch molecules [3,53]. The changes in the rehydration ratio of EHD-dried apple slices under various drying conditions are given in Figure 11.

From Figure 11, EHD-dried apple slices of 3 mm had slightly lower rehydration values. This might be because the standard weight of the dried apple sample, 2.0 g, was taken as a fixed parameter in the rehydration experiments, resulting in a smaller number of slices at higher thicknesses and thereby a reduced area for water absorption. Another reason for the higher rehydration ratio in thinner slices could be the difference in porosity. Dinani et al. [11] reported that shorter drying results in a more porous structure and thus higher water absorption rates. There was no significant difference in the rehydration ratio of EHD and hot air-dried apple slices (Figure 11).

#### 3.2.4. Disintegration Index

The disintegration index (DI) is another important parameter determining the damage to the internal microstructure of the dried products. The electrical conductivity method was used to determine the degree of cell disintegration. Usually, it is used to measure the severity of the treatments, for example, electroporation in pulsed electric field processing [54]. The leaching out of cellular compartments because of cell disintegration results in higher electrical conductivity of solution [21]. The DI values from our experiment are provided in Figure 12.

In our experiments, the disintegration index of EHD-dried apple slices did not change in the range of studied factors (Figure 12). A slightly higher disintegration was observed after hot air drying, indicating more damage to the cellular structure. The DI in EHD-dried apple slices was 0.48 ± 0.02 compared to 0.69 ± 0.12 for hot air-dried slices.

#### 3.2.5. Total Polyphenol Content

Fruits are rich in polyphenols, which are known for their antioxidant activity and health-promoting properties [55]. The total polyphenol content (TPC) in apples was determined as milligram gallic acid equivalent per gram of dry weight using the Folin–Ciocalteau method as shown in Figure 13. The TPC of fresh apples was 2.8 mg GAE/g dry weight.

Figure 13 shows that there was no significant difference in the TPC among EHD treatments in the range of studied factors. The TPC was mostly retained in EHD drying but was slightly lower in hot air-dried slices. The average TPC in EHD and hot air-dried apple slices was 1.95 ± 0.19 and 1.4 ± 0.05 mg GAE/g dry weight, respectively. Some studies reported an increase in phenolic compounds after hot air drying [56]. A couple of possible reasons for this phenomenon have been mentioned, as follows: (1) the release of phenolics due to cellular rupture, or (2) producing phenolics as secondary oxidation products. However, the temperature in our study did not exceed 40 °C, which is below the threshold for thermal degradation reactions and the formation of subsequent products.

In summary, our experiments did not show the significant effect of environmental factors on product quality in the range of the experimental design. Table 3 summarizes the statistical ANOVA *p*-values showing the significance (*p* < 0.05) of the main and interaction effects of factors (relative humidity, external airflow, slice thickness, and slice load density) on the quality of EHD-dried apple slices.

The ANOVA table (Table 3) shows the significant effect of thickness on both area and volumetric shrinkage. This could be due to the faster drying of thinner slices, as discussed above. A significant interaction effect of thickness, load density, and airflow on the rehydration ratio could be explained by the major effect of thickness, which was mostly observable in the 3 mm apple slices. It is also important to mention that the increase in relative humidity prolonged drying time, and in this way indirectly affected volume shrinkage, disintegration index, rehydration ratio, and phenolic content (*p* < 0.05). However, multiple means comparison tests could not be performed on the given relative humidity conditions.

## 4. Conclusions

The effect of material properties, such as slice thickness and load density, and environmental conditions, such as relative humidity and external airflow, on the drying kinetics, specific energy consumption, and quality of EHD-dried apple slices were studied. The drying characteristics and quality attributes were compared with hot air drying at 40 °C and 0.15 m/s as the control. The drying flux in the constant drying rate period mostly depended on load density and relative humidity. The drying flux substantially increased from 0.06 ± 0.01 g/m^2^s to 0.17 ± 0.02 g/m^2^s when the relative humidity decreased from 80% to 40%. The drying intensity in the falling drying rate period depended on the slice thickness. Effective diffusivity, as well as the drying rate constant, both increased with increases in air velocity in the range from 0 to 1.0 m/s. The smallest specific energy consumption was at 100% load density. The SEC in EHD drying was 10–12 times lower than in hot air drying.

The effect of electrohydrodynamic drying on product quality was not significant. No significant difference in color, disintegration index, and total polyphenolic content among EHD treatments was observed in the range of factors studied. In general, EHD-dried apple slices retained better quality compared to hot air drying, especially in color and shrinkage. The color change (ΔE) in EHD drying was 1.53 ± 0.4 compared to 6.78 ± 1.05 for hot air drying. The microstructural damage and degradation of phenolic compounds were also slightly lower in EHD drying. The total polyphenolic content in EHD drying was 1.95 ± 0.19 mg GAE/g dry weight versus 1.4 ± 0.05 mg GAE/g in hot air drying, while fresh apples had 2.8 mg GAE/g. Hence, EHD drying proved to be effective for thin fruit slices with better drying kinetics, low energy consumption, and superior quality. The impact of EHD drying on product texture and sensory properties should be further investigated.

## Figures and Tables

**Figure 1 foods-11-02765-f001:**
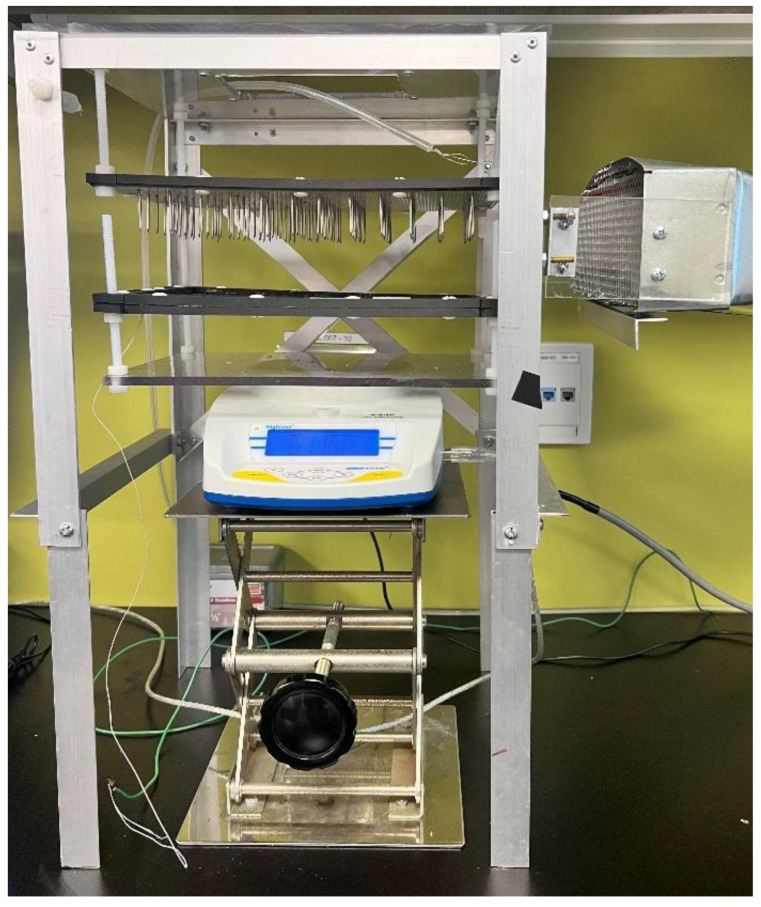
Electrode arrangement in EHD dryer.

**Figure 2 foods-11-02765-f002:**
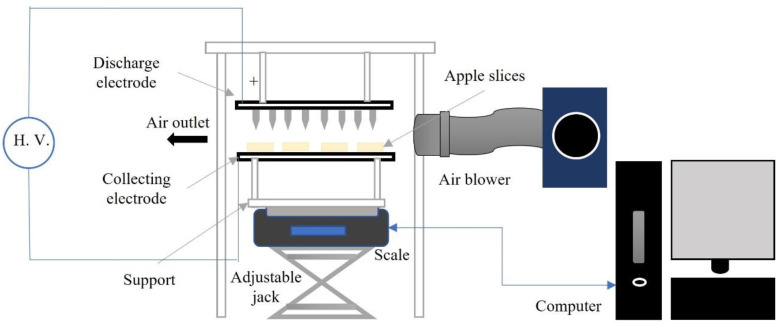
A schematic diagram of the EHD drying setup (H.V.–High Voltage).

**Figure 3 foods-11-02765-f003:**
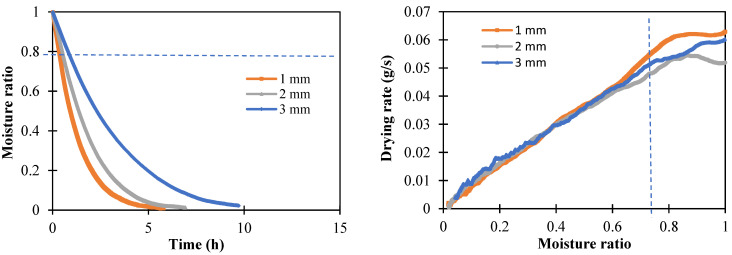
Drying kinetics of apple slices of different thicknesses (100% load—0.5 m/s—52% RH).

**Figure 4 foods-11-02765-f004:**
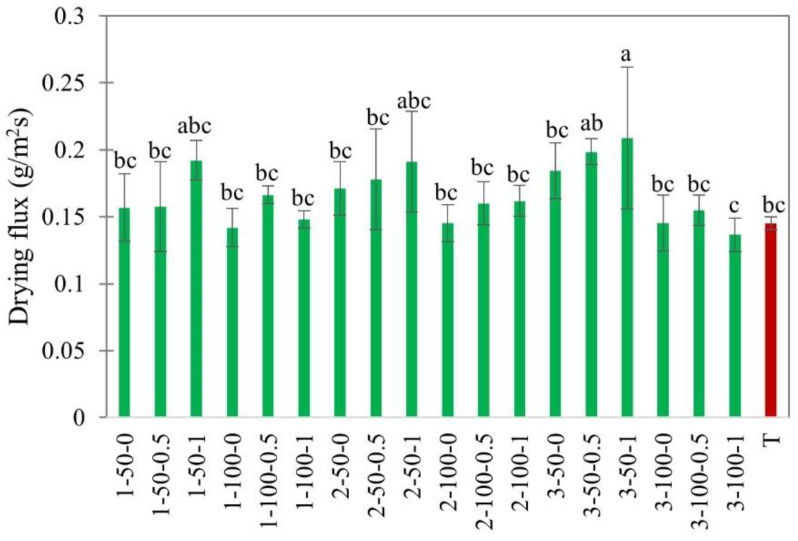
Drying flux of EHD (marked in green and labeled as thickness-slice load density-airflow) and hot air-dried (marked in red as T) apple slices. Different lower-case letters indicate statistically significant differences (*p* < 0.05).

**Figure 5 foods-11-02765-f005:**
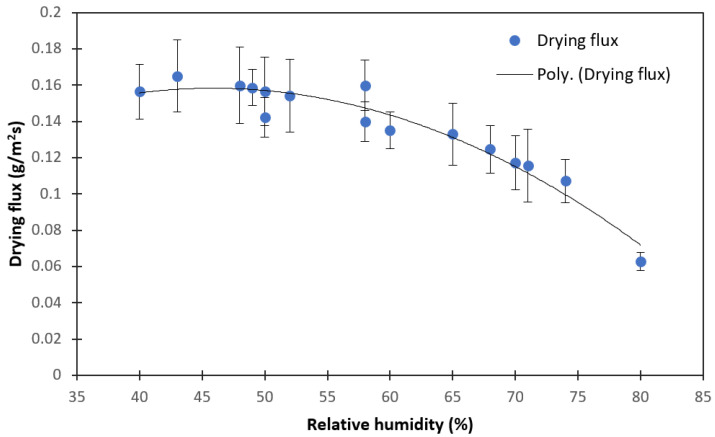
Changes in drying flux with relative humidity.

**Figure 6 foods-11-02765-f006:**
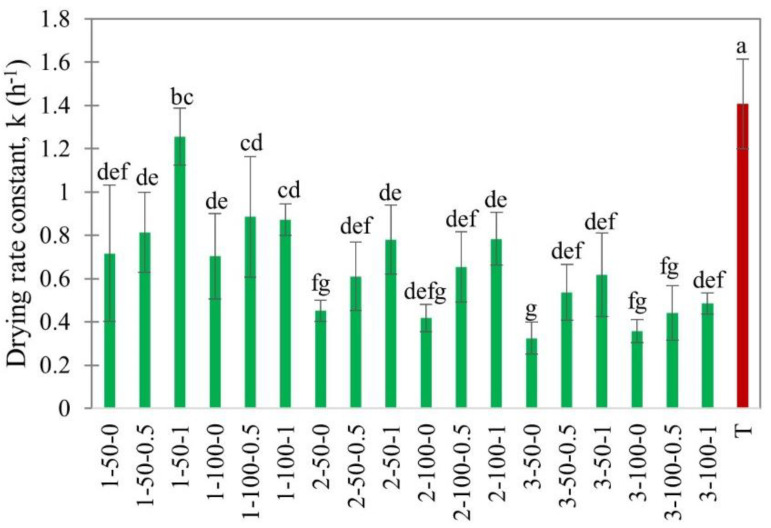
Drying rate constant of EHD (marked in green and labeled as thickness-slice load density-airflow) and hot air-dried (marked in red as T) apple slices. Different lower-case letters indicate statistically significant differences (*p* < 0.05).

**Figure 7 foods-11-02765-f007:**
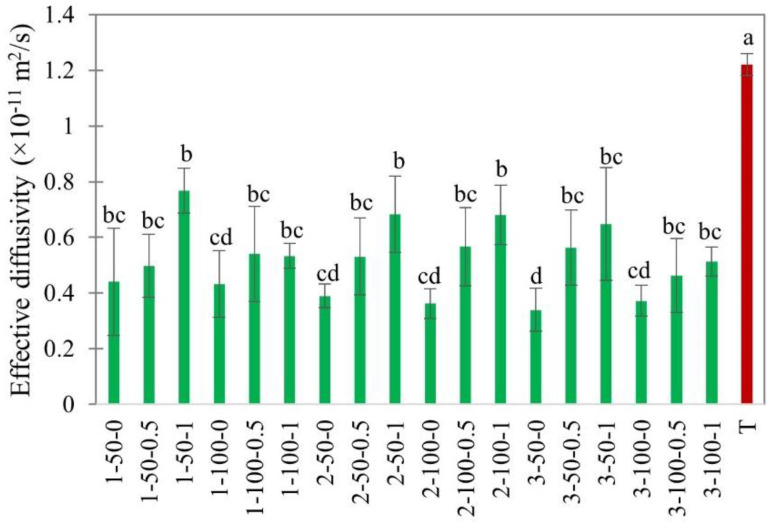
Effective diffusivity of EHD (marked in green and labeled as thickness-slice load density-airflow) and hot air-dried (marked in red as T) apple slices. Different lower-case letters indicate statistically significant differences (*p* < 0.05).

**Figure 8 foods-11-02765-f008:**
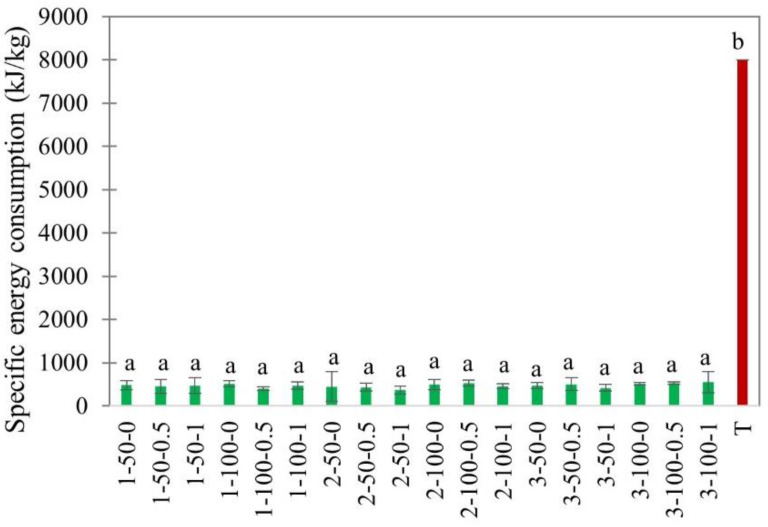
Specific energy consumption of EHD (marked in green and labeled as thickness-slice load density-airflow) and hot air-dried (marked in red as T) apple slices. Different lower-case letters indicate statistically significant differences (*p* < 0.05).

**Figure 9 foods-11-02765-f009:**
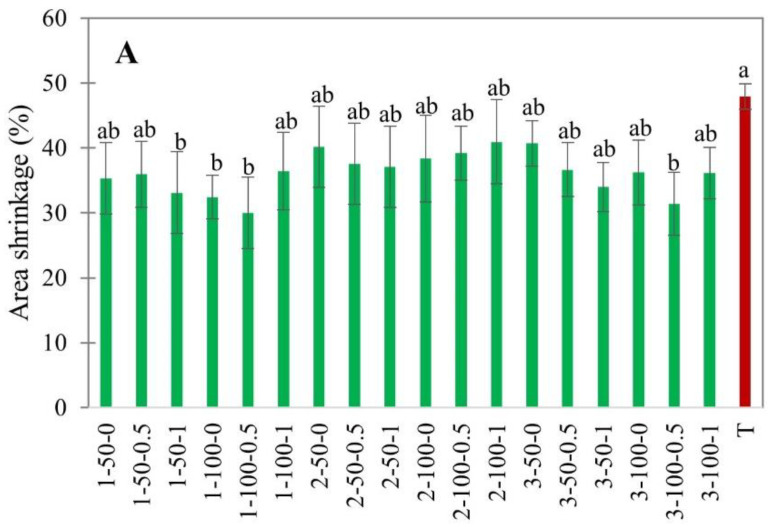
Area (**A**) and volumetric (**B**) shrinkage of EHD (marked in green and labeled as thickness-slice load density-airflow) and hot air-dried (marked in red as T) apple slices. Different lower-case letters indicate statistically significant differences (*p* < 0.05).

**Figure 10 foods-11-02765-f010:**
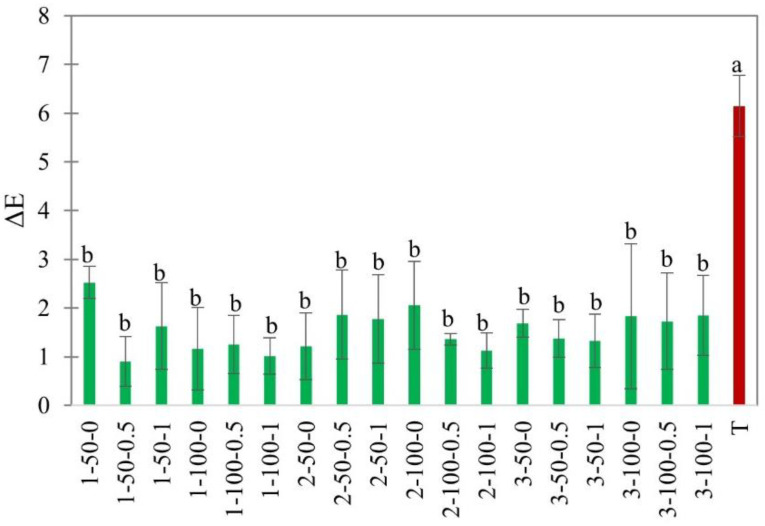
Color change, ΔE, of EHD (marked in green and labeled as thickness-slice load density-airflow) and hot air-dried (marked in red as T) apple slices. Different lower-case letters indicate statistically significant differences (*p* < 0.05).

**Figure 11 foods-11-02765-f011:**
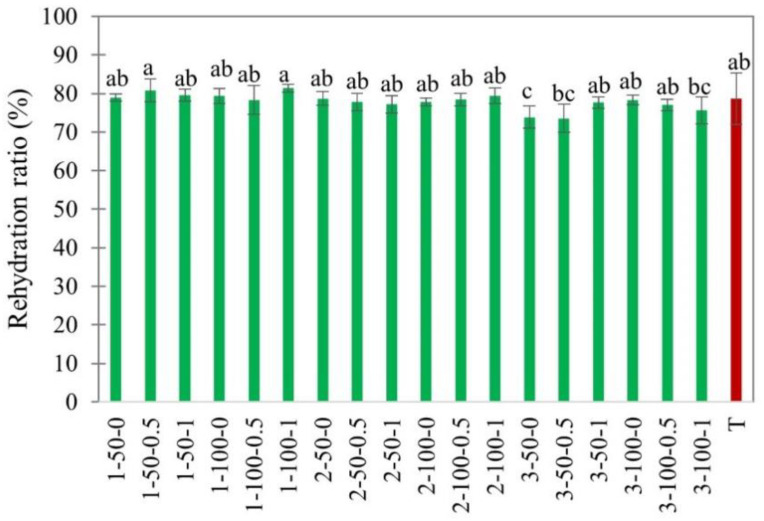
Rehydration ratio of EHD (marked in green and labeled as thickness-slice load density-airflow) and hot air-dried (marked in red as T) apple slices. Different lower-case letters indicate statistically significant differences (*p* < 0.05).

**Figure 12 foods-11-02765-f012:**
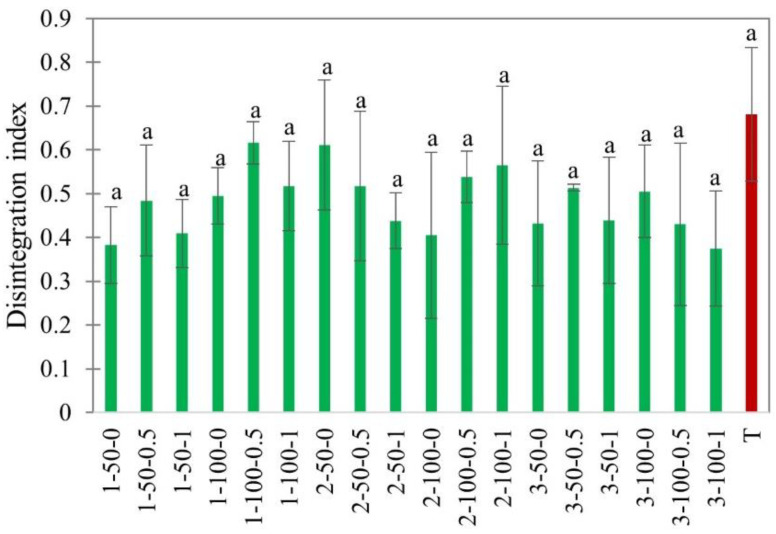
Disintegration Index of EHD (marked in green and labeled as thickness-slice load density-airflow) and hot air-dried (marked in red as T) apple slices. Different lower-case letters indicate statistically significant differences (*p* < 0.05).

**Figure 13 foods-11-02765-f013:**
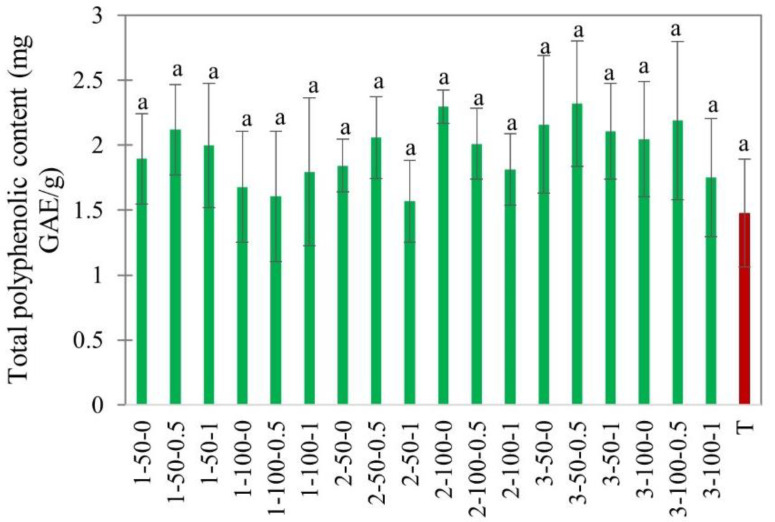
Total polyphenolic content of EHD (marked in green and labeled as thickness-slice load density-airflow) and hot air-dried (marked in red as T) apple slices. Different lower-case letters indicate statistically significant differences (*p* < 0.05).

**Table 1 foods-11-02765-t001:** Experimental design with three independent factors.

No.	Sample Code	Thickness (mm)	Load Density (%)	Airflow (m/s)
1	1-50-0	1	50	0
2	1-50-0.5	1	50	0.5
3	1-50-1	1	50	1
4	1-100-0	1	100	0
5	1-100-0.5	1	100	0.5
6	1-100-1	1	100	1
7	2-50-0	2	50	0
8	2-50-0.5	2	50	0.5
9	2-50-1	2	50	1
10	2-100-0	2	100	0
11	2-100-0.5	2	100	0.5
12	2-100-1	2	100	1
13	3-50-0	3	50	0
14	3-50-0.5	3	50	0.5
15	3-50-1	3	50	1
16	3-100-0	3	100	0
17	3-100-0.5	3	100	0.5
18	3-100-1	3	100	1

**Table 2 foods-11-02765-t002:** The *p*-values obtained from the multifactorial ANOVA using the general linear model at a 0.05 significance level for drying characteristics.

Factors	Drying Flux (g/m^2^s)	k (h^−1^)	Deff (×10^−11^ m^2^/s)	SEC (kJ/kg)
Thickness	0.206	**0.000**	0.218	0.212
Load	0.000	0.123	0.123	**0.031**
Airflow	0.041	**0.000**	**0.000**	0.445
Thickness*load	**0.018**	0.433	0.494	0.154
Thickness*airflow	0.896	0.549	0.710	0.290
Load*airflow	**0.041**	0.084	0.132	0.614
Thickness*load*airflow	0.614	0.181	0.336	0.738
Lack-of-Fit	0.311	0.504	0.844	0.132

Note: Significant effects are marked in bold.

**Table 3 foods-11-02765-t003:** The *p*-values obtained from the multifactorial ANOVA using the general linear model at a 0.05 significance level for quality attributes.

Factors	Area Shrinkage	Volume Shrinkage	ΔE	Rehydration Ratio	Disintegration Index	Total Polyphenolic Content
Thickness	**0.031**	**0.000**	0.547	0.000	0.256	0.108
Load	0.493	0.055	0.540	0.088	0.436	0.304
Airflow	0.544	0.967	0.216	0.388	0.267	0.171
Thickness*load	0.561	0.235	0.105	0.202	0.116	0.061
Thickness*airflow	0.879	0.197	0.654	0.806	0.751	0.451
Load*airflow	0.161	0.644	0.736	0.740	0.709	0.513
Thickness*load*airflow	0.934	0.325	0.042	**0.003**	0.140	0.831
Lack-of-Fit	0.897	0.729	0.359	0.497	0.679	0.124

Note: Significant effects are marked in bold.

## Data Availability

Data is contained within the article.

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
