# Peer review of "The Effect of Material Thickness, Load Density, External Airflow, and Relative Humidity on the Drying Efficiency and Quality of EHD-Dried Apples"

_foods, 2022, doi:10.3390/foods11182765_

Round 1

Reviewer 1 Report

The submitted article is comprehensive; the authors have developed a lab-scale EHD dryer and the experiment was performed over five months. In my opinion, it can be some minor corrections:

Page 1, lines 16-17, check units

Page 1, line 30, are there any other disadvantages of EHD rather than limited industrial applications?

Pages 1-2,4, Lines 24-25, 74-82, 118-120 and 134-140 check font size

Page 4, line 132, add initial moisture content of apple samples

Pages 10-11, in my opinion, figure 4, should be split into four different figures, so the results discussions are below the figure for each drying characteristic, it is a bit hard to follow the discussion of the results going back and forward. The same comment goes for Figure 6.

References are not prepared according to the journal guidance

Author Response

Reviewer 1

The submitted article is comprehensive; the authors have developed a lab-scale EHD dryer, and the experiment was performed over five months. In my opinion, it can be some minor corrections:

Thank you for your valuable comments and suggestions.

Page 1, lines 16-17, check units

Corrected

Page 1, line 30, are there any other disadvantages of EHD rather than limited industrial applications?

One of the main limitations of EHD drying is it is suitable only for the thin layer of materials, which is studied in this experiment and mentioned in lines 34-35.

Pages 1-2,4, Lines 24-25, 74-82, 118-120 and 134-140 check font size

All document was changed into 12 font size

Page 4, line 132, add initial moisture content of apple samples

The initial moisture content was added in the methodology

Pages 10-11, in my opinion, figure 4, should be split into four different figures, so the results discussions are below the figure for each drying characteristic, it is a bit hard to follow the discussion of the results going back and forward. The same comment goes for Figure 6.

Thank you for your suggestion. The figures were modified as per the comments.

Reviewer 2 Report

The new drying methods are very interesting and necessary. However, this manuscript needs some corrections.

Please provide more results in your abstract. Please correctly write the units in the abstract.

Introduction:

L24: please move (EHD) after the word "drying",

L25: please replace the word "dewatering" with "dehydration"

L41, L45, L47, L48: The authors cite the literature ([8, 9, 10, 6, 11, 12]) but do not specify what raw material the research was about.

In L55 authors write: "It was widely reported that EHD dried products retained the natural quality better than hot-air dried." but does not relate to specific raw materials.

Please indicate in picture 2 where the dried product is placed.

L129: please specify the storage temperature

How were the parameters in Equations 1 and 2 determined.

In equation 11 no "+" sign

At Figure 5 please add error bars.

Please consider adding the literature discussions in the chapters: 3.1.2. Drying rate constant, Effective diffusivity, Specific energy consumption.

In conclusions should include quantitative results.

There is a lot of mess in the numbering of chapters.

Author Response

Reviewer 2

The new drying methods are very interesting and necessary. However, this manuscript needs some corrections.

Thank you for your comments.

Please provide more results in your abstract. Please correctly write the units in the abstract.

The abstract was modified with more results.

Introduction: L24: please move (EHD) after the word "drying".

The abbreviation EHD is generally used for the word “ElectroHydroDynamic” in physics. Hence, we use EHD drying here.

L25: please replace the word "dewatering" with "dehydration

The word “dewatering” was replaced with “dehydration”.

L41, L45, L47, L48: The authors cite the literature ([8, 9, 10, 6, 11, 12]) but do not specify what raw material the research was about. In L55 authors write: "It was widely reported that EHD dried products retained the natural quality better than hot-air dried." but does not relate to specific raw materials.

Changes were made to the introduction as per the Reviewer's comments. The changes are shown in red using track changes.

Please indicate in picture 2 where the dried product is placed.

The figure was replaced after labelling apple slices in the picture

L129: please specify the storage temperature

The storage temperature was specified

How were the parameters in Equations 1 and 2 determined.

The explanations were added under the equations 1 and 2

In equation 11 no "+" sign

+ sign was added to the equation

At Figure 5 please add error bars.

Error bars were added to the figure

Please consider adding the literature discussions in the chapters: 3.1.2. Drying rate constant, Effective diffusivity, Specific energy consumption.

More literature discussions were added as per the comments.

In conclusions should include quantitative results.

Quantitative results were added to the conclusion

There is a lot of mess in the numbering of chapters.

Rechecked